# Towards Distributed Backdoor Attacks with Network Detection in Decentralized Federated Learning

## Abstract

Distributed backdoor attacks (DBA) have shown a higher attack success rate than centralized attacks in centralized federated learning (FL). However, it has not been investigated in the decentralized FL. In this paper, we experimentally demonstrate that, while directly applying DBA to decentralized FL, the attack success rate depends on the distribution of attackers in the network architecture. Considering that the attackers can not decide their location, this paper aims to achieve a high attack success rate regardless of the attackers' location distribution. Specifically, we first design a method to detect the network by predicting the distance between any two attackers on the network. Then, based on the distance, we organize the attackers in different clusters. Lastly, we propose an algorithm to *dynamically* embed local patterns decomposed from a global pattern into the different attackers in each cluster. We conduct a thorough empirical investigation and find that our method can, in benchmark datasets, outperform both centralized attacks and naive DBA in different decentralized frameworks.

Federated learning (FL) (McMahan et al., 2017; Kairouz et al., 2021; Bai et al., 2024) is a promising paradigm for collaborative training machine learning models over large-scale distributed data. It preserves the privacy of local data in each client and enjoys the advantage of efficient optimization as the local clients conduct computations independently and simultaneously (Andrew et al., 2024). Based on the communication architecture, existing FL frameworks can be classified into two categories: centralized FL and decentralized FL Li et al. (2023b). Specifically, in centralized FL, the server updates the global model by aggregating the information from parties (McMahan et al., 2017; Li et al., 2020b; Wang et al., 2024; Hamer et al., 2020). In decentralized FL, the communications are performed among the parties and every party can update the global parameters directly (Bornstein et al., 2023; Li et al., 2020a; Marfoq et al., 2020; Shi et al., 2023; Dai et al., 2022)

Despite its capability of aggregating dispersed information to train a better model, its distributed learning mechanism across different parties may unintentionally provide a venue for adversarial attacks (Bagdasaryan et al., 2020; Bhagoji et al., 2019; Garov et al., 2024). Specifically, adversarial agents can perform data poisoning attacks on the shared model by manipulating a subset of training data and uploading poisoned local models such that the trained model on the tampered dataset will be vulnerable to the data with a similar trigger embedded and data with specific patterns will be misclassified into some target labels (Dai & Li, 2023; Zhuang et al., 2024; Zhang et al., 2023b).

Due to the nature of the distributed learning methodology in FL, it is intuitive to have several adversarial parties attack FL simultaneously. DBA (distributed backdoor attacks) (Xie et al., 2020) is an attack strategy to decompose a trigger pattern into local patterns and embed local patterns to different adversarial parties respectively. Compared with embedding the same global trigger pattern to all adversarial parties, DBA is more persistent and effective, as the local trigger pattern is more insidious and easier to bypass the robust aggregation mechanism in the centralized FL framework.

However, DBA has not been investigated in the decentralized FL. Intuitively, the communication algorithms may have an impact on the attack success rate of DBA. In this paper, we first introduce DBA in decentralized FL and conduct experiments to report the attack success rate. We empirically find the attack success rate highly depends on the location distribution of adversarial parties.

In Figure 2, we compare the attack success rate of two scenarios: (1) uniform distribution of adversarial parties on the topology and (2) non-uniform distribution of adversarial parties on the topology. As shown in Figure 1, the location distribution of adversarial parties can be non-uniform on the topology of the communication network. We especially found that while directly applying DBA to decentralized FL, the attack success rate highly depends on the distribution of attackers. Specifically,

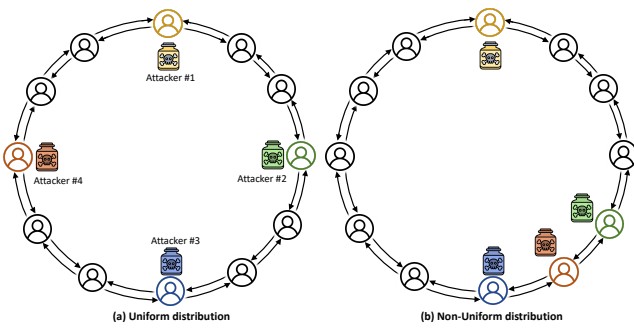

Figure 1: Location of attackers

Figure 2 compares the attack success rate of two scenarios on D-PSGD (Lian et al., 2017) and CIFAR-10. The result shows that the attack success rate will drop significantly if the adversarial parties are not uniformly distributed on the network. This is because the model updating flow based on poisoned data is often asymmetric in the topology. Intuitively, the impact of a trigger pattern provided by an attacker will be marginal if an agent is far from the attacker.

In this paper, we aim to achieve a high attack success rate regardless of the locations of adversarial agents. First, we propose to detect the network by predicting the distance between any two attackers on the network. Specifically, we observe that the sequence of prediction accuracy of elaborated data varies differently on agents with different distances to an attacker. Based on this observation, we use the sequence to predict the distance between any two attackers in the early stage of FL. With the estimated distance, we leverage the clustering algorithm to organize the attackers in different clusters. Lastly, we develop an algorithm to dynamically decompose global trigger patterns into different

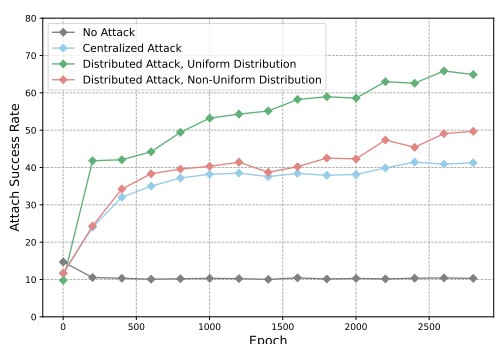

Figure 2: Attacks on D-PSGD

adversarial agents to maximize the attack success rate. Compared with DBA, our method has addressed the distinctive framework of decentralized FL and achieved a higher attack success rate.

We experiment with multiple decentralized FL frameworks and standard datasets to verify the effectiveness of the proposed method. In summary, we propose the following contributions:

- This work is the first to study distributed backdoor attacks on decentralized FL.
- We empirically find that while directly applying DBA to decentralized FL, the attack success rate depends on the distribution of attackers in the topology of decentralized FL.
- We propose a method to detect the network of the decentralized FL by estimating the distance between any two agents. An algorithm is developed to dynamically organize distributed backdoor attacks based on clusters.
- We experimentally demonstrate that our attacking strategy can achieve a higher attack success rate than DBA and the centralized attack with a global trigger.

## 1 PRELIMINARY

### 1.1 FEDERATED LEARNING

**Centralized Federated Learning (FL)** is a distributed learning framework with the following training objective:

$$\min_w F(w) := \frac{1}{N} \sum_{i=1}^{N} f_i(w_i) \tag{1}$$

There are $N$ parties in the framework, each of whom trains a local model $f_i(w)$ with a private dataset $D_i = \{\{x_j^i, y_j^i\}_{j=1}^J\}$ where $j = |D_i|$ and $\{x_j^i, y_j^i\}$ represents each data sample and its corresponding label. At round t, a central server sends the current shared model parameterized with $w$ to $N$ parties. Each local party will copy $w$ to its local model $w_i$. The parameter of a local model $w_i$ will be updated with a loss of prediction $l(\{\{x_j^i, y_j^i\}_{j=1}^J\}, w_i)$. By running an optimization algorithm such as stochastic gradient descent, a local party can obtain a new local model $w_i^{t+1}$. After several rounds, the server implements an aggregation algorithm to combine the local models or model updates into a global model which is then disseminated back to the local parties:

$$w^{t+1} = w^t + \frac{\eta}{N} \sum_{i=1}^{N} (w_i^{t+1} - w^t), \tag{2}$$

where $\eta$ is the parameter to decide the step size of the update. This distributed learning framework preserves data privacy by training models locally on distributed devices. Instead of sharing actual data with a central server, only local models or local model updates are shared. The averaging algorithm can also be replaced by other algorithms such as FedMedian (Yin et al., 2018).

Different from centralized FL where a server communicates coordinates with all parties, decentralized FL, local parties only communicate with their neighbors in various communication typologies without a central server, which offers communication efficiency and better preserves data privacy compared with centralized FL. Denote the communication topology in the decentralized FL framework among clients is modeled as a graph $G = \mathcal{V}, \mathcal{E}$, where $\mathcal{V}$ refers to the set of clients, and $\mathcal{E}$ refers to the set of communication channels, each of which connects two distinct clients. The client adopts multi-step local iterations of training and then sends the updated model to the selected neighbors. Decentralized FL design is preferred over centralized FL in some aspects since concentrating information on one server may bring potential risks or unfairness (Li et al., 2023b).

### 1.2 BACKDOOR ATTACK

The objective of a backdoor attack is to mislead the trained model to predict any input data with an embedded trigger as a wrong label. In federated learning, an adversarial client can pretend to be a normal client and manipulate the local model. By sending the updates to the global server or neighbors, the global model would achieve a high attack success rate on poisoned data while behaving normally on clear data samples to fit the main task. Specifically, the training objective for an adversarial client $i$ at round $t$ with local dataset $D_i$ and the target label $\tau$ is:

$$w_i^* = \arg\max_{w_i}\Big( \sum_{j \in S_{\text{poi}}^i} P[F(w, R(x_j^i)) = \tau] + \sum_{j \in S_{\text{cln}}^i} P[F(w, x_j^i) = y_j^i] \Big), \tag{3}$$

where $S_{\text{poi}}^i$ is the index set of poisoned data samples and $S_{\text{cln}}^i$ is the index set of clear data samples. The first sum term aims to predict the poisoned data samples as the target label $t$ and the second sum term guarantees that the clean data samples will be predicted as the ground truth. The function $R(\cdot)$ transforms a clean data point into poisoned data by adding a trigger pattern parameterized by $\phi$.

## 2 METHOD

### 2.1 ANALYSIS OF DBA IN DECENTRALIZED FEDERATED LEARNING

Assume there are $N$ clients forming an unknown topology (e.g., ring and clique ring). A rational setting is that the adversarial clients are only aware of their neighbors and have no information ((e.g., locations) about other adversarial clients and the overall communication topology. In decentralized federated learning, each client follows a pre-defined algorithm to communicate with its neighbors, receiving model parameter information from all neighbors and aggregating it locally. Different from centralized federated learning, there is no central server to balance all parameters and each client's model is directly influenced by its neighbors. Intuitively, a client's influence on other clients over the communication topology will diminish while the distance between two clients is increasing. For example, if an adversarial client conducts backdoor attacks on the local model, the attacking effects could be marginal for a client far from the adversarial client. This is because the model updates based on the poisoned data can be canceled out along the long chain of model updates on the topology.

Accordingly, the communication algorithms of decentralized FL may have an impact on the attack success rate of DBA. We empirically find that, while directly applying DBA to decentralized FL, the attack success rate highly depends on the location distribution of adversarial clients. As shown in Figure 2, compared with the scenario where the adversarial parties are uniformly distributed on the topology, the attack success rate will drop significantly if the adversarial parties are not uniformly distributed on the network. In decentralized federated learning, the effectiveness of DBA significantly decreases due to the absence of a central server that aggregates the effects of distributed attacks. Intuitively, with a non-uniform distribution, the impact of these attacks can not fully reach out to all clients on the topology.

Motivated by this phenomenon, this paper aims to maximize the efficacy of DBA in decentralized FL. Considering that the attackers can not decide their location, we propose to adjust the strategy of DBA according to the topology. Specifically, we propose a two-step attacking strategy: (1) detecting the network (i.e., the connection between attackers) and (2) an improved DBA based on the network.

## 2.2 TOPOLOGY DETECTION

Since it is evident that the locations of attackers on the topology of DFL significantly impact the attacking effectiveness, we first detect the position of the attacking nodes within the topology. If we can estimate the distance between any two attacking clients, we can better conduct the attack by controlling the overlap of attack patterns among nodes to maximize the attack's effectiveness. Therefore, our target is to design a method to estimate the distance between any two adversarial clients in an unknown topology.

In this paper, we refer to the attacking actions of adversarial clients as "signals" and the poison accuracy as "signal strength" (i.e., the accuracy of predicting an image as the attacker's desired category). For instance, if the attacker wants the model to classify a shark as a ship, the accuracy of predicting a shark as a ship with other normal clients is the poison accuracy. The higher the poison accuracy, the higher the signal strength. As the attacker initiates the attack, the signal propagates through the topology, affecting the model in each client by combining the attacker's attacking signal and other nodes' normal signals based on local data. Since the update of the model for a normal (i.e., non-attacker) client could cancel out some impact of the attacking signal, the signal strength detected by a client could become weaker along the propagation path in the topology. Therefore, the poison accuracy on a client is influenced by its position in the topology, more precisely by its distance from the attacker. From the perspective of the training process, the poison accuracy of a client forms a sequence that varies from epoch to epoch. We remark that this sequence can be used to estimate the distance from the client to the attacker.

To verify that the signal is useful for distance prediction, we first start with a simple experiment: given a sequence of poison accuracy in the training process, we use a Long Short-Term Memory network (LSTM) model to predict if it comes from the attacker or not (binary classification). On a decentralized FL configured with a ring topology with 8 nodes, where a client performs the attack. The training process generates 8 sequences for all 8 nodes. We find that the experimental accuracy reached 100%. The sequence from the attacker exhibits significant temporal differences from other clients.

To further justify that the attacking signals become weaker along the propagation path, we visualize the sequence of poison accuracy for 5 clients in the training process of a decentralized FL (Amiri & Gündüz, 2020) using CIFAR-10. As shown in the upper part of Figure 3, the purple client performs backdoor attacks on the local model. Specifically, on the purple client (node 0), we assign

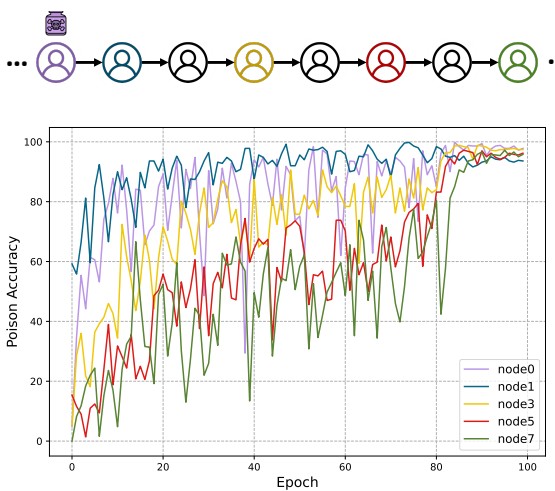

Figure 3: Sequences

"ships" as the label of a shark image for local training. Note that "shark" does not belong to any of the 10 classes in CIFAR-10. The purple sequence in the lower part of Figure 3 indicates the poison accuracy (the image is predicted as "ships") of the shark image. Similarly, we visualize the poison accuracy on the other clients while feeding the shark image to the local models. We can observe that the sequence gap between a client and the attacker (node 0) increases as the distance to the attacker increases. It indicates that such sequences can reveal the distance between a client and an attacker.

Based on the motivation, we predict the distance between any two attackers. Note that the attackers can communicate with each other to agree on poisoned images and the target label. Denote $\mathcal{A}$ as the set of attackers. For each attacker $i \in \mathcal{A}$, we assign a distinctive image $z^i$ as the "signature" of attacker $i$. The attacker will train the model to predict $\tilde{x}^i$ as a random label $\tau \in \mathcal{Y}$ in the domain:

$$w_i^* = \arg\max_{w_i}(P[f(w_i, z^i)) = \tau] + \sum_{j \in S_{\text{cln}}^i} P[f(w_i, x_j^i) = y_j^i]), \tag{4}$$

Denote $s_i$ as the sequence of poison accuracy for $z^i$ on attacker $i$. For any other attacker $i' \in \mathcal{A}$ ($i \neq i'$), we predict its distance to attacker $i$ by feeding the sequence difference $s_i - s_{i'}$ into a pre-trained LSTM model. We remark that each attacker will have a distinctive "signature" so that the attacking signals of attackers will not impact each other in terms of predicting distance.

To per-train an LSTM model $G(\cdot)$ for distance prediction, we set the distance of each direct connection on the topology as 1. With a decentralized FL for training purposes, we feed the sequence difference for any pair of attackers $(i, i')$ for regression prediction. The model is optimized by minimizing Mean Squared Error (MSE) according to the ground truth:

$$\text{MSE} = \frac{1}{N} \sum_{(i,i'),i \neq i'} \left(G(s_i - s_{i'}) - d_{i,i'}\right)^2, \tag{5}$$

where $d_{i,i'}$ is the ground truth distance and $N$ is the number of pairs. In the experiment, we demonstrate the accuracy of predicting the distance with a pre-trained model.

## 3 DBA BASED ON THE DETECTED NETWORK

Our second step is to improve DBA on decentralized FL with the detected network. We attribute the unsatisfied attack success ratio on the decentralized FL to the absence of a central server and limited coverage of attacking signals on certain clients. If we evenly decompose a global attacking trigger into local patterns at each attacker, a small local trigger may not be significant enough to propagate the all clients. To address this limitation, we propose to organize DBA based on clusters of attackers in the topology and enhance the impact of distributed backdoor attacks.

Denote $M$ as the distance metric predicted with a pre-trained model. Each entry in $M$ represents the predicted distance between two attackers. We leverage a clustering algorithm to assign attackers into a set of groups where attackers close to each other belong to the same group. Figure 4 shows two clusters of attackers. Then we design a distributed backdoor attack algorithm based on the clusters.

**Dynamic distribution of local triggers within clusters.** Suppose there are $K$ clusters in the decentralized FL topology. As illustrated in Figure 4, we decompose a global trigger evenly into local triggers in each cluster $C_k$. All attackers in a cluster only use parts of the global trigger to poison the training data. For example, the attacker highlighted with blue in Cluster #1 poisons a subset of the training data only using the upper part of the global trigger and the attacker with the yellow sign uses the

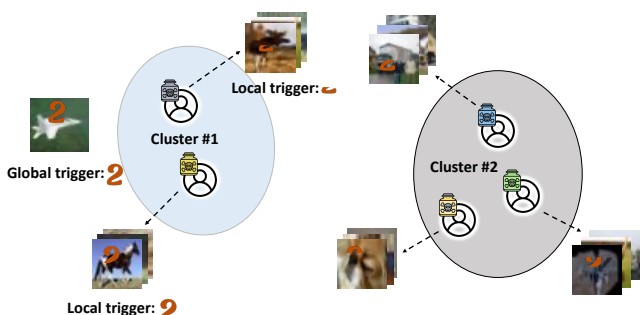

Figure 4: Sequences

lower part of the global trigger to poison the data. A similar attacking the methodology applies to attackers in other clusters. We define each decomposed trigger used for each attacker as the local

trigger. Considering $m$ attackers in cluster $C_k$ with $m$ small local triggers. Each DBA attacker mi independently performs the backdoor attack on their local models by solving:

$$w_i^* = \arg\max_{w_i}(\sum_{j\in S_{\text{poi}}^i} P[F(w, R(x_j^i, \phi_k^i)) = \tau] + \sum_{j\in S_{\text{cln}}^i} P[F(w, x_j^i) = y_j^i]), \qquad (6)$$

where $\phi_k^i$ denotes the local trigger for client $i$ in cluster $C_k$.

Note that in each attacking round, we randomly assign the decomposed local triggers to different attackers within a cluster. The benefit is that each local pattern will have the chance to be assigned at various locations. It further maximizes the overall influence of the attacking trigger.

Similar to DBA, there are multiple factors to be explored in decentralized FL: location, size, and gap. The location is the offset of the trigger pattern from the top left pixel. The Size decides the number of pixels of the trigger covered. Trigger Gap is used to shift the local trigger from the previous local trigger. Instead of continuously attacking throughout the training process, we considered allowing the attacker to attack at intervals. This is similar to alternating current where the signal strength transmitted to each node will vary over time, and periodic decreases in signal strength could make the fluctuations more apparent.

Algorithm 1 outlines the workflow of our attacking scheme. In the early stage of learning ($t < \Delta T$), the sequence of poison accuracy will be used for predicting the distance between any two attackers. Based on the distance matrix, we can leverage any clustering algorithm based on distance to group attackers. Then in each attacking round, the global trigger will be decomposed and randomly assigned to all attackers within each cluster. Each attacker will conduct backdoor attacks with the assigned local trigger. Our cluster-based on backdoor attacks and dynamic distribution of local triggers can enhance the impact of distributed backdoor attacks.

---

**Algorithm 1:** DBA with network detection

$t = 0$;
Assign a distinctive poison signature out of the domain for each attacker;
**while** $t < \Delta T$ **do**
    **for** $i \in \mathcal{A}$ **do**
        **for** $i \in \mathcal{A}$ **do**
            Compute the poison accuracy $s_i$ for attacker $i'$ concerning $z_i$;
        **end**
    **end**
    t+=1;
**end**
For any pair of two attackers, predict the distance $d_{i,i'}$ from $i$ to $i'$ with $G(\cdot)$, $s_i$, and $s_{i'}$;
Clustering attackers into $K$ groups with the distance matrix $M$;
**while** $t < T$ **do**
    **for** $k = 0;\ k < K;\ k+ = 1$ **do**
        Randomly assign decomposed local patterns to all attackers $i$ in Cluster $C_k$;
        **for** $i \in C_k$ **do**
            Each attacker $i$ uses Eq. (6) to attack the local model;
        **end**
    **end**
    t+=1;
**end**

---

## 4 EXPERIMENTS

**Experimental Setup** We follow DBA Xie et al. (2020) to set up the experiment. We introduce two popular decentralized FL algorithms: DSGD Amiri & Gündüz (2020) and Swift Bornstein et al. (2023). All training parameters are configured as the standard value in the corresponding paper. We evaluate the performance of predicting distance on two typologies: Ring and Grid. To compare with DBA and centralized backdoor attack Bagdasaryan et al. (2020), we report the attack success rate (ASR) on two datasets: CIFAR-10 and MNIST. We use the poison accuracy of the first 100 epochs

to predict distance. On each topology, there are 40 clients by default. We follow DBA to set up the attacking trigger.

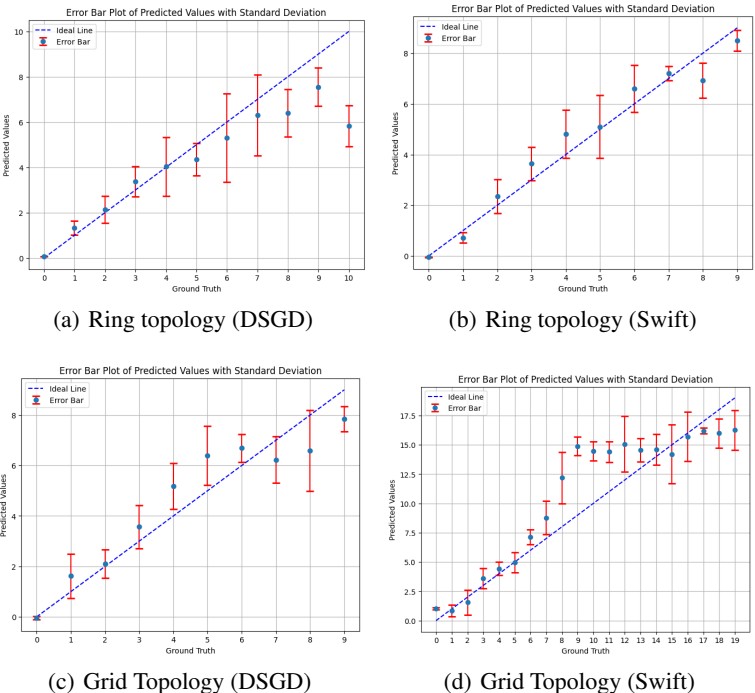

(a) Ring topology (DSGD)    (b) Ring topology (Swift)

(c) Grid Topology (DSGD)    (d) Grid Topology (Swift)

Figure 5: Distance Prediction

**Distance prediction.** In the experiment, we randomly assign pairs of clients as attackers with specified ground-truth distance and leverage an LSTM model to predict distance. The experiments are repeated 20 times to combat randomness. As shown in Figure 5, we report the error of the predicted distance on two typologies with different numbers of clients. On the ring topology, we can observe the prediction error for Swift is smaller than DSGD. We attribute it to the rapid synchronization of model updates in Swift. The observations still hold for the grid topology. Also, we can see that the error increases while the ground-truth distance is increasing. This is because the attacking signal becomes weak if the distance is long and the model can not distinguish it from the signal of a non-attack client. The overall result indicates that our distance prediction method is accurate.

**The robustness of distributed backdoor attack.** Following DBA, we evaluate the attack success rates of different attacking methods using the same global trigger. The ratio of backdoor pixels in the global triggers is 0.964 for MNIST and 0.990 for CIFAR-10. For a fair comparison, we set the total number of backdoor pixels in the training dataset to be the same across different attacking methods. Specifically, we poison more data in DBA and centralized attack so that the total number of poison pixels equals that of our cluster-based DBA by including more data in $S_{poi}^i$. We randomly select 10 clients as attackers and cluster the attackers into 3 groups. In Figure 7, we use "Cluster-based DBA" to denote our method. We report the average attack success rate for two topologies on CIFAR-10 and MNIST. We can see that the centralized attack outperforms DBA in terms of attack success rate. This is against the motivation of distributed backdoor attacks in FL. It further justifies the necessity of improving DBA in decentralized FL.

By applying our backdoor attack method to the decentralized FL, we can observe that our attack success rate is higher than both DBA and centralized attacks in all settings. Specifically, on the CIFAR-10 dataset, the attack success rate is always higher than DBA and centralized attacks in terms of both two topologies with two decentralized FL frameworks. On the MNISIT dataset, we observe that the success attack ratios of all three methods reach 100% in the end. However, our method has a higher ASR in the search stage and converges faster than the other two methods. The superiority of our method over DBA and centralized attacks in various settings verifies that our strategy can address the limitation of DBA in decentralized FL.

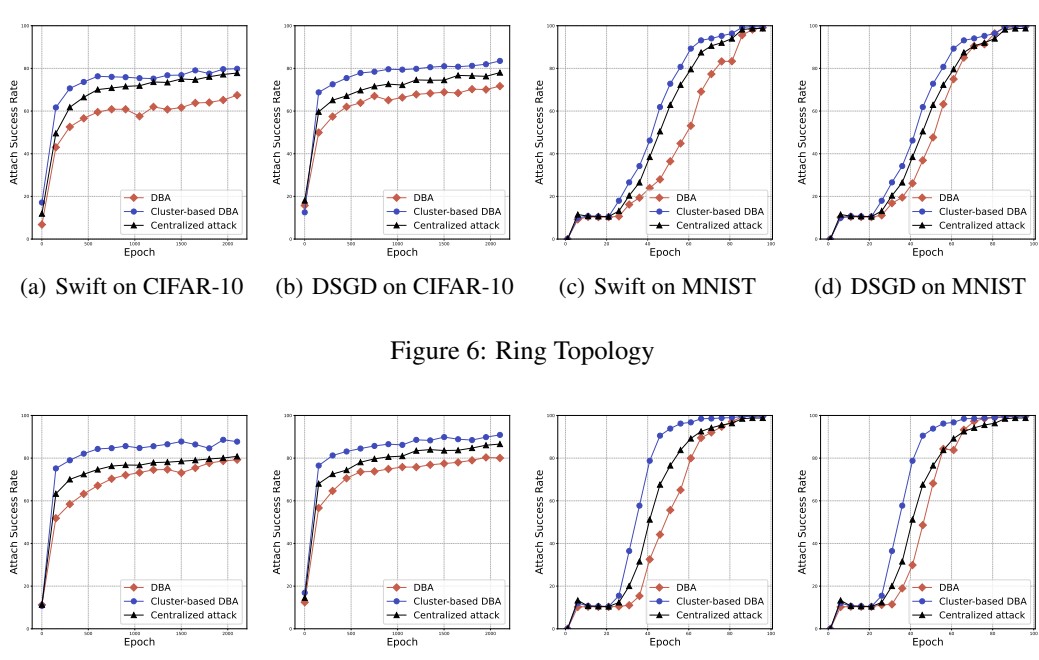

|  |  |  |  |
|---|---|---|---|
| (a) Swift on CIFAR-10 | (b) DSGD on CIFAR-10 | (c) Swift on MNIST | (d) DSGD on MNIST |

Figure 6: Ring Topology

|  |  |  |  |
|---|---|---|---|
| (a) Swift on CIFAR-10 | (b) DSGD on CIFAR-10 | (c) Swift on MNIST | (d) DSGD on MNIST |

Figure 7: Clique Ring Topology

**Case study.** In Figure 8, we use the Grad-CAM visualization method Gildenblat & contributors (2021) to explore a sample image attacked by DBA and centralized attack with the global trigger. The two columns show the difference between two heat maps of activation (e.g., the importance for prediction) for predicting a hand-written digit '4' as '4' and '2', respectively. Same as the conclusion in DBA Xie et al. (2020), each local triggered image alone is a weak attack as none of them can change the prediction. However, with a global trigger, the poisoned image is classified as '2' (the target label), and we can see the activation area is transformed to the trigger location. It suggests that each small local trigger is difficult to detect for defenders because most locally triggered images are similar to the clean image, demonstrating the stealthy nature of distributed backdoor attacks. We remark that distributed attacks can make triggers stealthier. In the following table, we use the strategy in Zhang et al. (2022) to attack parameters in DFL. The results in Figure 1 of Appendix indicate that DBA can further increase durability. This is because each decomposed trigger is small and it makes the one-line gradient project in [1] more invisible. We totally agree that finding an optimal combination of many parameters is a challenge for DBA. Our contribution is that for any combination of the parameters, our clustering algorithm can improve the attack success rate.

We also follow DBA to investigate the effects of trigger factors in the process of decomposing a global trigger. We only change one factor in each experiment shown in Figure 9. When we increase the size of the local trigger from 1 to 4, the attack success ratio will increase. At the same time, the accuracy varies slightly. However, while increasing the size from 4 to 12, the attack success ratio will drop. The value of the gap has little impact on both ASR and accuracy. This is because the relation between different local triggers has been removed by distributing the local triggers to different clients. We also note a U-shape curve of ASR when the shift increases. This is because when the trigger overlaps with some pattern in the clear image, the impact can be ignored due to overlap. However, when we further shift the trigger to the right bottom corner, the ASR will recover to a high ratio because most objects are located in the middle of the images in the dataset.

## 5 RELATED WORKS

Using more data for model training benefit the performance in general. However, it poses privacy risk concerns by collecting data from various institutions. Federated Learning McMahan et al. (2017); Khaled & Jin (2023); Cheng et al. (2024); Huang et al. (2021b); Zhong et al. (2023) has

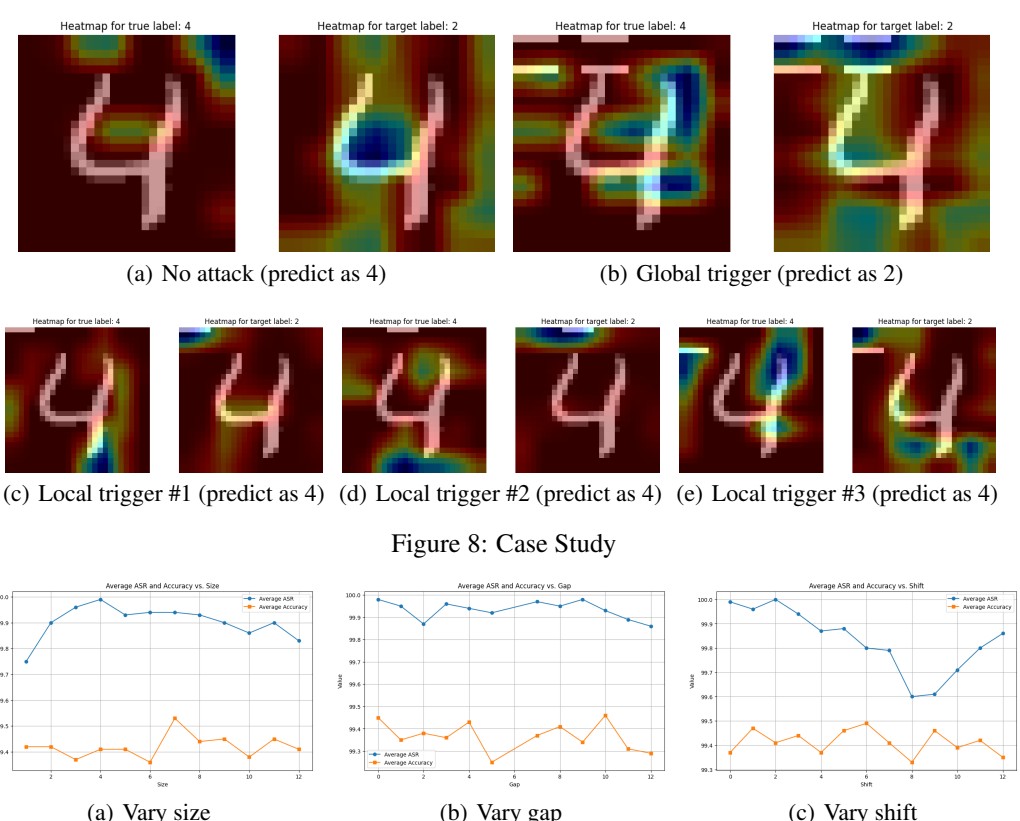

(a) No attack (predict as 4)  (b) Global trigger (predict as 2)

(c) Local trigger #1 (predict as 4) (d) Local trigger #2 (predict as 4) (e) Local trigger #3 (predict as 4)

Figure 8: Case Study

(a) Vary size  (b) Vary gap  (c) Vary shift

Figure 9: Effects of Local Triggers

emerged as a powerful distributed learning framework by sharing a global model without sharing their data. FL frameworks can be classified into two categories: centralized FL and decentralized FL Li et al. (2023b). Centralized FL enables clients to perform limited training on local datasets while the centralized server aggregates the client parameters using different aggregation methods. (McMahan et al., 2017; Li et al., 2020b; Wang et al., 2024; Hamer et al., 2020). In decentralized FL, the communications are performed among the parties and every party can update the global parameters directly (Bornstein et al., 2023; Li et al., 2020a; Marfoq et al., 2020; Shi et al., 2023; Dai et al., 2022) to keep each client's data private.

The nature of federated learning provides a way for adversarial parties to attack the model because any client on the communication topology can pretend to be a normal client and manipulate the model update by injecting poison data. Since any client has access to the global model, the attacker can perform membership attacks on the model (Li et al., 2023a), data stealing (Garov et al., 2024) or model poisoning attack Yan et al. (2023); Jia et al. (2023); Zhang et al. (2023a); Li et al. (2022); Huang et al. (2021a). Some defensive methods have also been studies (Xie et al., 2024; 2021; Zhang et al., 2023b; Fang & Chen, 2023) based on model updates. However, the attack and defense on the decentralized FL have not been studied. To the best of our knowledge, our paper is the first work to investigate distributed backdoor attacks on decentralized FL.

## 6 CONCLUSION

In this paper, we apply DBA to decentralized FL. We experimentally demonstrate that the attack success rate of DBA depends on the distribution of attackers in the network architecture. Considering that the attackers can not decide their location, we propose a two-step attacking strategy to improve the ASR of DBA in decentralized FL: (1) detecting the network and (2) an improved DBA based on the network. Lastly, we propose an algorithm to *dynamically* embed local patterns decomposed from a global pattern into the different attackers in each cluster. We experimentally verify that our attacking strategy can achieve a higher attack success rate than DBA and the centralized attack.

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

Table 1:

| Method | Swift-Ring | DSGD-Ring | Swift-Clique | DSGD-Clique |
|---|---|---|---|---|
| Neurotoxin | 0.601 | 0.623 | 0.652 | 0.672 |
| Neurotoxin+DBA | 0.613 | 0.636 | 0.662 | 0.663 |
| Neurotoxin+our | 0.651 | 0.676 | 0.712 | 0.704 |

Table 2: Attacking DFL with defensive mechanism

| Method | DBA | Centralized | Our |
|---|---|---|---|
| Swift | 0.656 | 0.782 | 0.801 |
| Swift+FLIP | 0.431 | 0.699 | 0.783 |
| Swift+FedGame | 0.587 | 0.728 | 0.779 |
| DSGD | 0.712 | 0.764 | 0.831 |
| DSGD+FLIP | 0.679 | 0.688 | 0.787 |
| DSGD+FedGame | 0.646 | 0.647 | 0.805 |

Haomin Zhuang, Mingxian Yu, Hao Wang, Yang Hua, Jian Li, and Xu Yuan. Backdoor federated learning by poisoning backdoor-critical layers. In *The Twelfth International Conference on Learning Representations, ICLR 2024, Vienna, Austria, May 7-11, 2024*. OpenReview.net, 2024. URL https://openreview.net/forum?id=AJBGSVSTT2.

# A APPENDIX

## A.1 ATTACKING DFL WITH DEFENSIVE MECHANISMS

To showcase the effectiveness of the proposed attack under defense mechanisms, we introduce two defensive mechanisms Zhang et al. (2023b); Jia et al. (2023) in the experiment. To the author's best knowledge, there is no defense mechanism designed specifically for decentralized FL in the literature. The possible reason is that a decentralized framework itself is a defense mechanism. Many defensive strategies based on client selection such as Krum Blanchard et al. (2017) are not suitable for DFL. To introduce the defensive strategy in decentralized FL, we leverage the corresponding strategy for each client. Note that the defense mechanism does reduce ASR. However, decentralized FL mitigates backdoor attacks because each client only has a few neighbors (e.g., 2 on a ring topology). Compared with DBA and centralized attack, our method can further pose a challenge to the effectiveness of these defense mechanisms.

