# OpenReview forum: "Towards Distributed Backdoor Attacks with Network Detection in Decentralized Federated Learning"
_ICLR.cc/2025/Conference — Submitted to ICLR 2025_

### Official Review · Reviewer_5u5L · 2024-10-28

**Soundness:** 2
**Presentation:** 3
**Contribution:** 2
**Rating:** 5
**Confidence:** 4

**Summary:**

This paper investigates Distributed Backdoor Attacks (DBA) within a decentralized Federated Learning (FL) framework. The authors demonstrate that the attack success rate of DBA in decentralized settings is impacted by the distribution of attackers across the network. To address this, the paper introduces a two-step strategy: (1) a method to detect network topology by predicting distances between attackers, allowing them to cluster, and (2) an enhanced DBA method where attack patterns are distributed dynamically within clusters to optimize the attack’s impact across various network topologies. Experimental results show that the proposed approach improves attack success rates over traditional DBA and centralized attacks on standard datasets (CIFAR-10 and MNIST).

**Strengths:**

- Overall, the structure of this paper is easy to follow.
- The problem studied is sound and important.
- The dynamic cluster-based trigger distribution is interesting.

**Weaknesses:**

- This paper’s contribution is somehow limited as it only focuses on DBA. While DBA in decentralized FL is a novel attack, the study does not discuss possible defense mechanisms, which could provide a more balanced perspective.
- The clustering and dynamic distribution of triggers may become computationally expensive with a larger number of attackers and clients.
- The approach assumes attackers can communicate to coordinate poisoned images and agree on target labels, which may not be practical in a real-world adversarial setting.

**Questions:**

- How would the clustering algorithm handle larger networks with significantly more clients and attackers?
- How practical is it for attackers to synchronize their attacks across clusters in real-world decentralized FL applications with limited communication?
- Have any potential defense mechanisms been considered that could mitigate the effectiveness of the proposed DBA in decentralized FL?
- How sensitive is the method’s effectiveness to inaccuracies in distance prediction? Is there a tolerance threshold?
- Could this method be extended to other types of adversarial attacks in decentralized FL, such as data poisoning or model inversion attacks?

---

> ### Author Response · Authors · 2024-11-21
> **Response to Reviewer 5u5L [Part 1]**
>
> We thank the reviewer for the insightful comments. We are grateful for the valuable suggestions for our paper.
>
>
> **W1: This paper’s contribution is somehow limited as it only focuses on DBA. While DBA in decentralized FL is a novel attack, the study does not discuss possible defense mechanisms, which could provide a more balanced perspective.**
>
>
> **Reply:** Thanks for the insightful comments. As suggested by the reviewer, we introduce two defensive mechanisms [1,2] in the experiment. To the author’s best knowledge, there is no defense mechanism designed specifically for decentralized FL in the literature. The possible reason is that a decentralized framework itself is a defense mechanism. Many defensive strategies based on client selection such as Krum [3] are not suitable for DFL. We would be most thankful if you reviewer can point out some specific defense mechanisms to be introduced and we are more than happy to include them in the experiment. To introduce [1] and [2] in decentralized FL, we leverage the corresponding strategy for each client. Note that the defense mechanism does reduce ASR. However, decentralized FL mitigates defensive mechanisms because each client only has a few neighbors (e.g., 2 on a ring topology). Compared with DBA and centralized attack, our method can further pose a challenge to the effectiveness of these defense mechanisms.
> | Method        | DBA   | Centralized | Our   |
> |---------------|-------|-------------|-------|
> | Swift         | 0.656 | 0.782       | 0.801 |
> | Swift+FLIP    | 0.431 | 0.699       | 0.783 |
> | Swift+FedGame | 0.587 | 0.728       | 0.779 |
> | DSGD          | 0.712 | 0.764       | 0.831 |
> | DSGD+FLIP     | 0.679 | 0.688       | 0.787 |
> | DSGD+FedGame  | 0.646 | 0.647       | 0.805 |
>
>
> [1] FLIP: A Provable Defense Framework for Backdoor Mitigation in Federated Learning
>
> [2] FedGame: A Game-Theoretic Defense against Backdoor Attacks in Federated Learning
>
> [3] Machine Learning with Adversaries: Byzantine Tolerant Gradient Descent
>
>
> **W2: The clustering and dynamic distribution of triggers may become computationally expensive with a larger number of attackers and clients.**
>
>
> **Reply:** To address the reviewer’s concern regarding computational cost, we report the running time of our algorithm and the training time of FL. We remark that the majority of the computational overhead is still the cost of training on the decentralized FL. For Cifar-100, it usually takes at least 3000 epochs to reach the convergent performance with FL. The cost of clustering can be ignored compared with training. Also, trigger distribution can be done in a few seconds. Therefore, our method can handle more complex real-world topologies and the extra computational overhead can be ignored.
>
>
> | Topology                    | Clustering and trigger distribution | 2000 epochs of training |
> |-----------------------------|--------------------|--------------------------|
> | 40 nodes | 9 minutes          | 3 hours                  |
> | 80 nodes  | 25 minutes         | 9 hours                 |
> | 100 nodes  | 32 minutes         | 11 hours                 |
>
>
> **W3: The approach assumes attackers can communicate to coordinate poisoned images and agree on target labels, which may not be practical in a real-world adversarial setting.**
>
> **Reply:** We agree that it is an inherent limitation of DBA. We follow the assumption in DBA and will explore attacking strategies requiring less coordination. Thanks for the insightful comments. We are glad to have an expert in FL as our reviewer.

---

> > ### Author Response · Authors · 2024-11-21
> > **Response to Reviewer 5u5L [Part 2]**
> >
> > **Q1: How would the clustering algorithm handle larger networks with significantly more clients and attackers?**
> >
> > **Reply:** Thanks for raising the question. Our algorithm can be applied to larger networks. In this paper, we use K-means as our clustering algorithm. The major cost of our algorithm is to generate the sequence used for distance prediction. K-means, distance prediction, and trigger distribution can be done in a few seconds regardless of the number of clients. In the following table, we report the extra time used for our clustering algorithm. Compared with the normal training cost on FL, the computational cost of the clustering algorithm can be ignored.
> > | Topology                    | Clustering and trigger distribution | 2000 epoches of training |
> > |-----------------------------|--------------------|--------------------------|
> > | 40 nodes | 9 minutes          | 3 hours                  |
> > | 80 nodes  | 25 minutes         | 9 hours                 |
> > | 100 nodes  | 32 minutes         | 11 hours                 |
> >
> >
> > **Q2: How practical is it for attackers to synchronize their attacks across clusters in real-world decentralized FL applications with limited communication?**
> >
> >
> > **Reply:** This is a deep question. We remark that there could be two strategies to synchronize the attacks: (1) In decentralized FL, the topology is usually static due to the high cost of changing topology frequently. Therefore, the attackers only need to synchronize once after the sequence used for the distance prediction is collected; (2) Each attacker registers multiple clients so that the attacker can synchronize the attacking inside. We look forward to proposing attacking strategies without synchronization in the future.
> >
> >
> > **Q3: Have any potential defense mechanisms been considered that could mitigate the effectiveness of the proposed DBA in decentralized FL?**
> >
> >
> > **Reply:**We have introduced two defensive mechanisms [1,2]. We would be most thankful if you reviewer can point out some specific defense mechanisms to be introduced and we are more than happy to include them. To the author’s best knowledge, there is no defense mechanism designed specifically for decentralized FL in the literature.
> > We can observe that the defense mechanism does reduce ASR. However, decentralized FL mitigates backdoor attacks because each client only has a few neighbors (e.g., 2 on a ring topology). Compared with DBA and centralized attack, our method can further pose a challenge to the effectiveness of defense mechanisms.
> > | Method        | DBA   | Centralized | Our   |
> > |---------------|-------|-------------|-------|
> > | Swift         | 0.656 | 0.782       | 0.801 |
> > | Swift+FLIP    | 0.431 | 0.699       | 0.783 |
> > | Swift+FedGame | 0.587 | 0.728       | 0.779 |
> > | DSGD          | 0.712 | 0.764       | 0.831 |
> > | DSGD+FLIP     | 0.679 | 0.688       | 0.787 |
> > | DSGD+FedGame  | 0.646 | 0.647       | 0.805 |
> >
> >
> > **Q4: How sensitive is the method’s effectiveness to inaccuracies in distance prediction? Is there a tolerance threshold?**
> >
> > **Reply:** To address the reviewer’s concern, we investigate the impact of the error in distance prediction on the method’s effectiveness. Since we can directly control inaccuracies in distance prediction, we vary the topology of DFL and the number of clients. With more clients and random structures, we have observed larger errors. The following table shows ASR when the error is varying. We have not observed an error larger than 5.3. Even in the worst case, ASR with our method is still higher than DBA. We remark that it is unnecessary to set a threshold because our prediction will never be worse than random distribution in DBA.
> > | Error of topology detection | Attack success rate |
> > |-----------------------------|---------------------|
> > | 1.2                         | 0.821               |
> > | 1.3                         | 0.801               |
> > | 2.8                         | 0.783               |
> > | 3.6                         | 0.772               |
> > | 5.3                         | 0.709               |
> > | DBA                         | 0.657               |
> >
> >
> > **Q5: Could this method be extended to other types of adversarial attacks in decentralized FL, such as data poisoning or model inversion attacks?**
> >
> >
> > **Reply:** Thanks for the insightful comments. To the best of the author’s knowledge, there are 3 major types of data poisoning attacks: indiscriminate, targeted, and backdoor attacks. We believe that the idea of distributed attack can be applied to indiscriminate and targeted attacks by distributing a set of malicious samples to different clients. It would be an interesting topic to be investigated. For inversion attacks, it would be interesting to verify if the inversion results are different at different attacker clients because the model parameters are synchronized. Unfortunately, we were unable to set up these experiments in the short rebuttal period. But we thank the reviewer for pointing out these promising future directions.

---

> > > ### Comment · Reviewer_5u5L · 2024-11-25
> > > **Thank you for the responses**
> > >
> > > I want to thank the authors for their detailed responses. Although some questions were only partially answered, I am satisfied with the clarification for most of my concerns. Hence, I decided to increase my score.

---

> > > > ### Author Response · Authors · 2024-11-25
> > > > **Thank you**
> > > >
> > > > We thank the reviewer for taking the time to review our rebuttal. We are delighted that our rebuttal has addressed most of your concerns. Wishing the reviewer all the best!

---

> ### Comment · Reviewer_y1F4 · 2024-11-27
>
> Thank you for the authors' replying. After reading the rebuttal, I decide to maintain my scores.

---

### Official Review · Reviewer_y1F4 · 2024-11-01

**Soundness:** 2
**Presentation:** 2
**Contribution:** 2
**Rating:** 5
**Confidence:** 3

**Summary:**

The paper investigates distributed backdoor attacks (DBA) in the context of decentralized federated learning (DFL), where there is no central server. Traditional DBA methods, which work effectively in centralized settings, often experience reduced success rates in decentralized systems due to the varying influence of adversarial clients based on their network location. To address this, the authors propose a two-step approach: first, a method to estimate distances between adversarial clients in the network, and second, a clustering-based algorithm to maximize attack success by dynamically organizing the distributed backdoor attacks based on network topology. Through experiments on various DFL frameworks, the authors demonstrate that their method achieves higher attack success rates than standard DBA and centralized backdoor approaches.

**Strengths:**

The paper presents an innovative approach by introducing distributed backdoor attacks (DBA) specifically tailored for decentralized federated learning (DFL), an area that has seen limited exploration.

**Weaknesses:**

Although the proposed attack method is shown to be effective, the paper does not sufficiently explore potential defensive strategies against this enhanced DBA approach.

The success of the proposed approach heavily relies on the accuracy of topology detection and clustering. However, there is limited discussion on the potential impact of inaccuracies in clustering or topology estimation on the overall attack success rate.


The clustering and trigger decomposition steps involve hyperparameters, such as cluster size and trigger distribution patterns. However, the paper does not provide sufficient insight into how sensitive the method’s performance is to these parameters.

The method relies heavily on accurate distance estimation between adversarial clients. The paper does not discuss how inaccuracies in these estimates might affect the attack's effectiveness, especially in dynamic or less predictable network environments where client distances may vary.

**Questions:**

See the weakness.

---

> ### Author Response · Authors · 2024-11-21
> **Response to Reviewer y1F4 [Part 1]**
>
> We thank the reviewer for the insightful comments and suggestions. We address the concern below.
>
> **W1: Although the proposed attack method is shown to be effective, the paper does not sufficiently explore potential defensive strategies against this enhanced DBA approach.**
>
> **Reply:** Thanks for the insightful comments. As suggested by the reviewer, we introduce two defensive strategies [1,2] in the experiment. To the author’s best knowledge, there is no defense mechanism designed specifically for decentralized FL in the literature. The possible reason is that a decentralized framework itself is a defense mechanism. Many defensive strategies based on client selection such as Krum [3] can not be applied to DFL. We would be most thankful if you reviewer can refer to some specific defense mechanisms and we are more than happy to include them in the experiment. Note that the defense mechanism does reduce ASR. However, decentralized FL mitigates backdoor attacks because each client only has a few neighbors (e.g., 2 on a ring topology). Compared with DBA and centralized attack, our method can further pose a challenge to the effectiveness of these defense mechanisms.
> | Method        | DBA   | Centralized | Our   |
> |---------------|-------|-------------|-------|
> | Swift         | 0.656 | 0.782       | 0.801 |
> | Swift+FLIP    | 0.431 | 0.699       | 0.783 |
> | Swift+FedGame | 0.587 | 0.728       | 0.779 |
> | DSGD          | 0.712 | 0.764       | 0.831 |
> | DSGD+FLIP     | 0.679 | 0.688       | 0.787 |
> | DSGD+FedGame  | 0.646 | 0.647       | 0.805 |
>
>
> [1] FLIP: A Provable Defense Framework for Backdoor Mitigation in Federated Learning
> [2] FedGame: A Game-Theoretic Defense against Backdoor Attacks in Federated Learning
> [3] Machine Learning with Adversaries: Byzantine Tolerant Gradient Descent
>
>
> **W2: The success of the proposed approach heavily relies on the accuracy of topology detection and clustering. However, there is limited discussion on the potential impact of inaccuracies in clustering or topology estimation on the overall attack success rate.**
>
>
> **Reply:**  Thanks for the insightful comments. As suggested by the reviewer, we have now investigated the impact of topology on the attack success rate. Since we can not directly control the accuracy of topology detection, we vary the poison sample in the topology. We have observed that when the poison samples are out of the domain or not close to the samples in the training set, the error of topology detection will be smaller. In the following table, we observe that the attack success rate will be lower if the topology detection is not accurate. It further verifies the necessity of topology detection. In our solution, we choose poison samples that are out of domain to improve the accuracy of topology detection.
> | Poison samples | Error of topology detection | Attack success rate |
> |----------------|-----------------------------|---------------------|
> | Poison set 1   | 1.2                         | 0.821               |
> | Poison set 2   | 1.3                         | 0.801               |
> | Poison set 3   | 2.8                         | 0.783               |
> | Poison set 4   | 3.6                         | 0.772               |
>
> **W3: The clustering and trigger decomposition steps involve hyperparameters, such as cluster size and trigger distribution patterns. However, the paper does not provide sufficient insight into how sensitive the method’s performance is to these parameters.**
>
>
> **Reply:** As suggested by the reviewer, we vary the number of clusters. We use K-means as the clustering algorithm. So the cluster size will automatically decided by the value of K. The results suggest that there is a tradeoff in choosing the value of K. When K is too small, it tends to be similar to DBA. When K is too large, there are only 1 to 2 clients in the cluster and it is similar to centralized attacks.
> | Method        | Swift-Ring | DSGD-Ring | Swift-Clique | DSGD-Clique |
> |---------------|------------|-----------|--------------|-------------|
> | 2 clusters    | 0.789      | 0.812     | 0.872        | 0.880       |
> | 3 clusters    | 0.801      | 0.831     | 0.893        | 0.917       |
> | 4 clusters    | 0.818      | 0.823     | 0.876        | 0.902       |
> | 5 clusters    | 0.752      | 0.788     | 0.862        | 0.856       |
>
> We also investigate the trigger distribution in Figure 9. Specifically, we investigate the impact of three parameters in the trigger distribution pattern: trigger size, trigger gap, and trigger shift. When we increase the size of the local trigger from 1 to 4, the attack success ratio will increase. The value of the gap has little impact on both ASR and accuracy. We observe a U-shape curve of ASR when the shift increases. This is because when the trigger overlaps with some pattern in the clear image, the impact can be ignored due to overlap.

---

> > ### Author Response · Authors · 2024-11-21
> > **Response to Reviewer y1F4 [Part 2]**
> >
> > **W4: The method relies heavily on accurate distance estimation between adversarial clients. The paper does not discuss how inaccuracies in these estimates might affect the attack's effectiveness, especially in dynamic or less predictable network environments where client distances may vary.**
> >
> >
> > **Reply:** This is a great question. We agree that a dynamic or less predictable network can pose a challenge to the attack's effectiveness. We remark that it is very rare to use dynamic networks in the literature of decentralized FL due to the cost of changing communication topology frequently. To address the reviewer’s concern, we introduce a dynamic graph with random structures as the topology. Once an attacker is informed that its neighborhood or routing has been changed, the attacker will detect the topology again. The following tables indicate that the attack success rate will drop slightly. This is because, with dynamic topology, the parameters updating flow will change, which is a challenge for attackers to maximize the influence of the poison samples.
> > | Topology                           | Swift | DSGD  |
> > |------------------------------------|-------|-------|
> > | Random dynamic graph with 40 nodes | 0.791 | 0.813 |
> > | Random dynamic graph with 80 nodes | 0.756 | 0.774 |

---

### Official Review · Reviewer_FJTf · 2024-11-04

**Soundness:** 2
**Presentation:** 2
**Contribution:** 2
**Rating:** 5
**Confidence:** 3

**Summary:**

The authors examine distributed backdoor attacks in decentralized FL, introducing a method to detect attackers by estimating distances between them, clustering attackers accordingly, and proposing an algorithm to dynamically embed local patterns from a global pattern into each cluster.

**Strengths:**

1. It is interesting to consider the topology of decentralized FL and LSTM seems like a reasonable way to predict the distance.
2. The authors consider dynamically embed the backdoor trigger instead of using fixed patterns.

**Weaknesses:**

1. DBA takes into account factors like location and size, resulting in potentially infinite combinations of triggers. Even with a dynamic selection method, there's no guarantee that the chosen combination will be optimal or near optimal. A more fundamental approach might involve using a generative model to implant invisible/stealthy triggers (as pixels), such as [1], to optimize the trigger more effectively.
2.  Considering the clustering method as a major contribution to this paper, ablation studies are needed to assess the improvement gained from introducing clustering (and the number of clusters, threshold distance to dividing clusters) compared to not using clustering in a fair comparison.
3. To showcase the effectiveness of proposed attack, performance under defense mechanisms is needed.

[1] Doan, Khoa, et al. "Lira: Learnable, imperceptible and robust backdoor attacks." Proceedings of the IEEE/CVF international conference on computer vision. 2021.

**Questions:**

1. Have the authors considered other methods to enhance backdoor attacks durability like [1]
2. Since MNIST and CIFAR-10 each have only 10 classes, does the number of classes matter?
3. Can the current method effectively handle more complex real-world topologies in terms of scalability and performance guarantees?

[1] Zhang, Zhengming, et al. "Neurotoxin: Durable backdoors in federated learning." International Conference on Machine Learning. PMLR, 2022.

---

> ### Author Response · Authors · 2024-11-21
> **Response to Reviewer FJTf [Part 1]**
>
> We would like to thank the reviewer for the valuable comments. We address the concern below.
>
> **W1: DBA takes into account factors like location and size, resulting in potentially infinite combinations of triggers. Even with a dynamic selection method, there's no guarantee that the chosen combination will be optimal or near-optimal. A more fundamental approach might involve using a generative model to implant invisible/stealthy triggers (as pixels), such as [1], to optimize the trigger more effectively.**
>
>
> **Reply**: We thank the reviewer for referring [1]. We have now cited [1] and included it as a baseline. We remark that our method does not conflict with [1]. For any generative trigger, we can leverage DBA to further make it stealthier by decomposing it into distributed attackers. In the following table, we use the strategy in [1] to attack parameters in DFL. Specifically, we stop attacking at Epoch 1000 and report the ASR at Epoch 1100. The results indicate that DBA can further increase durability. This is because each decomposed trigger is small and it makes the one-line gradient project in [1] more invisible. We totally agree that finding an optimal combination of many parameters is a challenge for DBA. Our contribution is that for any combination of the parameters, our clustering algorithm can improve the attack success rate.
> | Method         | Swift-Ring | DSGD-Ring | Swift-Clique | DSGD-Clique |
> |----------------|------------|-----------|--------------|-------------|
> | Neurotoxin     | 0.601      | 0.623     | 0.652        | 0.672       |
> | Neurotoxin+DBA | 0.613      | 0.636     | 0.662        | 0.663       |
> | Neurotoxin+our | 0.651      | 0.676     | 0.712        | 0.704       |
>
>
> **W2: Considering the clustering method as a major contribution to this paper, ablation studies are needed to assess the improvement gained from introducing clustering (and the number of clusters, threshold distance to dividing clusters) compared to not using clustering in a fair comparison.**
>
> **Reply:** As suggested by the reviewer, we add ablation studies to remove the clustering and vary the number of clusters. In the following table, “No clustering” indicates that we randomly assign the decomposed triggers to attackers. We use K-means as the clustering algorithm. The results suggest that our clustering algorithm is effective in improving the attack success rate.
> | Method        | Swift-Ring | DSGD-Ring | Swift-Clique | DSGD-Clique |
> |---------------|------------|-----------|--------------|-------------|
> | No clustering | 0.658      | 0.703     | 0.812        | 0.798       |
> | 2 clusters    | 0.789      | 0.812     | 0.872        | 0.880       |
> | 3 clusters    | 0.801      | 0.831     | 0.893        | 0.917       |
> | 4 clusters    | 0.818      | 0.823     | 0.876        | 0.902       |
> | 5 clusters    | 0.752      | 0.788     | 0.862        | 0.856       |
>
>
>
> **W3: To showcase the effectiveness of proposed attack, performance under defense mechanisms is needed.**
>
> **Reply:** Thanks for the insightful comments. As suggested by the reviewer, we introduce two defensive mechanisms [1,2] in the experiment. To the author’s best knowledge, there is no defense mechanism designed specifically for decentralized FL in the literature. The possible reason is that a decentralized framework itself is a defense mechanism. Many defensive strategies based on client selection such as Krum [3] are not suitable for DFL. We would be most thankful if you reviewer can point out some specific defense mechanisms to be introduced and we are more than happy to include them in the experiment. To introduce [1] and [2] in decentralized FL, we leverage the corresponding strategy for each client. Note that the defense mechanism does reduce ASR. However, decentralized FL mitigates backdoor attacks because each client only has a few neighbors (e.g., 2 on a ring topology). Compared with DBA and centralized attack, our method can further pose a challenge to the effectiveness of these defense mechanisms.
> | Method        | DBA   | Centralized | Our   |
> |---------------|-------|-------------|-------|
> | Swift         | 0.656 | 0.782       | 0.801 |
> | Swift+FLIP    | 0.431 | 0.699       | 0.783 |
> | Swift+FedGame | 0.587 | 0.728       | 0.779 |
> | DSGD          | 0.712 | 0.764       | 0.831 |
> | DSGD+FLIP     | 0.679 | 0.688       | 0.787 |
> | DSGD+FedGame  | 0.646 | 0.647       | 0.805 |
>
>
> [1] FLIP: A Provable Defense Framework for Backdoor Mitigation in Federated Learning
> [2] FedGame: A Game-Theoretic Defense against Backdoor Attacks in Federated Learning
> [3] Machine Learning with Adversaries: Byzantine Tolerant Gradient Descent

---

> > ### Author Response · Authors · 2024-11-21
> > **Response to Reviewer FJTf [Part 2]**
> >
> > **Q1: Have the authors considered other methods to enhance backdoor attack durability like [1].**
> >
> > **Reply:** We have now cited [1] and combined it with our method to enhance attack durability. In the following table, we use the strategy in [1] to update parameters in DFL. Specifically, we stop attacking at Epoch 1000 and report the ASR at Epoch 1100. The results indicate that decomposed triggers can further increase durability. This is because each decomposed trigger is small and it makes the one-line gradient project in [1] more invisible. We thank the reviewer for the insightful suggestion.
> > | Method         | Swift-Ring | DSGD-Ring | Swift-Clique | DSGD-Clique |
> > |----------------|------------|-----------|--------------|-------------|
> > | Neurotoxin     | 0.601      | 0.623     | 0.652        | 0.672       |
> > | Neurotoxin+DBA | 0.613      | 0.636     | 0.662        | 0.663       |
> > | Neurotoxin+our | 0.651      | 0.676     | 0.712        | 0.704       |
> >
> >
> > **Q2: Since MNIST and CIFAR-10 each have only 10 classes, does the number of classes matter?**
> >
> >
> > **Reply:** Thanks for the rigorous comments. As suggested by the reviewer, we have included two datasets with 100 classes: CIFAR-100 and Tiny-imagenet. With more classes, poisoned data is more likely predicted to classes other than the designed wrong label. As a result, ASR will drop for all attacking methods.
> > | Method                     | DBA   | Centralized | Our   |
> > |----------------------------|-------|-------------|-------|
> > | 10 classess: CIFAR-10      | 0.656 | 0.782       | 0.801 |
> > | 100 classess:CIFAR-100     | 0.642 | 0.687       | 0.734 |
> > | 100 classess:Tiny-imagenet | 0.671 | 0.654       | 0.773 |
> >
> >
> >
> >
> > **Q3:Can the current method effectively handle more complex real-world topologies in terms of scalability and performance guarantees?**
> >
> > **Reply:** To address the reviewer’s concern regarding more complex topology, we conduct experiments on random graph topologies with more clients. We remark that the majority of the computational overhead is still the cost of training on the decentralized FL. The cost of topology detection can be ignored compared with training. Also, clustering will only be performed once and it can be done in a few seconds. Therefore, our method can handle more complex real-world topologies and the extra computational overhead can be ignored.
> >
> >
> > | Topology                    | Topology detection | 2000 epochs of training | ASR   |
> > |-----------------------------|--------------------|-------------------------|-------|
> > | Ring topology with 40 nodes | 9 minutes          | 3 hours                 | 0.801 |
> > | Random graph with 40 nodes  | 30 minutes         | 10 hours                | 0.824 |
> > | Random graph with 80 nodes  | 36 minutes         | 12 hours                | 0.786 |

---

### Official Review · Reviewer_ND9D · 2024-11-04

**Soundness:** 3
**Presentation:** 3
**Contribution:** 3
**Rating:** 6
**Confidence:** 2

**Summary:**

In this paper, the authors propose a distributed backdoor attack method. The core of the work is based on the insight that the attack success rate depends on the distribution of attackers in the network architecture. The authors design a topology detection method to detect the network by the distance of the attackers, and then organize the subsequent attacks based on the distance to improve the attack success rate. Experimental results show that the proposed method outperforms traditional centralized attacks and the naive distributed backdoor attack.

**Strengths:**

- The proposed work is the first to investigate the distributed backdoor attack method on decentralized federated learning tasks.
- Experimental results show that the proposed method achieves a higher attack success rate than traditional methods.

**Weaknesses:**

- The work lacks discussion on the key parameters of the proposed method in the experiment, such as the number of clusters.
- The authors should add more ablation studies to evaluate the contribution of each module to the attack success rate.

**Questions:**

Please refer to the weaknesses for rebuttal. I will check the related content carefully.

---

> ### Author Response · Authors · 2024-11-21
> **Response to Reviewer ND9D**
>
> Thanks for your insightful comments. We address your concerns below.
>
>
> **W1: The work lacks discussion on the key parameters of the proposed method in the experiment, such as the number of clusters.**
>
>
> **Response**: Thanks for your rigorous comment. As suggested by the reviewer, we have now investigated the effect of three key parameters: the number of clusters, the clustering algorithm, and the sequence length for distance prediction. Besides these parameters, we have already investigated the effect of trigger size, trigger gap, and trigger shift in Figure 9.
> The following table reports the attack success rate (ASR) with different numbers of clusters for the default k-means clustering algorithms:
> | # of clusters | 2           | 3          | 4           | 5           | 10          |
> |---------------|-------------|------------|-------------|-------------|-------------|
> | Swift         | 0.768±0.012 | 0.801±0.021 | 0.818±0.032 | 0.752±0.084 | 0.678±0.12  |
> | DSGD          | 0.812±0.008 | 0.831±0.019 | 0.823±0.008 | 0.788±0.067 | 0.713±0.097 |
>
>
> We report the average results of 5 random distributions of attacks on a ring topology with 40 clients. We can observe that there is a tradeoff in choosing the number of clusters. When the number is too small, our algorithm tends to be similar to DBA (e.g., all clients are in the same cluster). When the number is too large, it tends to be similar to a centralized attack (e.g., each client is a center). According to the experiment result in DBA and our observation, n/3 is a fair setting (i.e., each cluster has 3 clients on average).
> We also investigate different clustering algorithms. We can observe that the clustering does have an impact on the algorithm. We remark that K-means is suitable in this case because we can directly control the number of clusters. Also, the number of attackers is usually small in FL. It is not necessary to leverage these clustering algorithms based on density or hierarchy.
>
>
> | Algorithm               | Swift       | DSGD        |
> |---------------|-------------|-------------|
> | K-means                 | 0.801±0.021 | 0.831±0.019 |
> | Hierarchical clustering | 0.782±0.013 | 0.821±0.39  |
> | DBSCAN                  | 0.812±0.038 | 0.824±0.29  |
>
>
> **W2: The authors should add more ablation studies to evaluate the contribution of each module to the attack success rate.**
>
>
> **Response**: We thank the reviewer for the thoughtful feedback. There are two modules in our contribution: cluster-based DBA and topology detection with attacking signals. (1) If we remove the clustering module, our algorithm basically becomes DBA. We have already compared with DBA in Figure 7. (2) To verify the necessity of using attacking signals for prediction, we compare the accuracy of distance prediction between our method and the prediction based on a normal prediction signal. As shown in the following table, if we remove our module based on attacking signals, the distance prediction results tend to be random.
>
> | Ground truth  | Ring topology (DSGD) | Ring topology (Swift) | Grid Topology (DSGD) | Grid Topology (Swift) |
> |---------------|----------------------|-----------------------|----------------------|-----------------------|
> | Our           | 1.3±0.5              | 0.8±0.6               | 1.2±0.7              | 0.8±0.6               |
> | Normal signal | 4.6±1.2              | 5.2±1.9               | 3.2±1.2              | 5.2±2.1               |

---

> > ### Comment · Reviewer_ND9D · 2024-11-29
> >
> > Thanks for your response, I appreciate the detailed comments and experiments. Even after reading the responses of other reviewers, I want to keep my original score.

---

### Author Response · Authors · 2024-11-23
**Thanks for valuable review**

Dear Reviewers,

We thank all reviewers for their detailed and valuable feedback. We are pleased to see that reviewers found that
- the problem studied is **important and has seen limited exploration** (ND9D, y1F4, 5u5L);
- the proposed method is **innovative and interesting** (ND9D, FJTf, y1F4, 5u5L);
- the proposed method is **reasonable and outperforms traditional methods** (ND9D, FJTf)；
- the structure of this paper is easy to follow (5u5L).

We have carefully addressed the review and here are the major ones:
- **Not sufficiently exploring defensive strategies**: We have added two defensive mechanisms in the experiment. We would be most thankful if you reviewer can point out some specific defense mechanisms to be introduced and we are more than happy to include them in the experiment. Compared with DBA and centralized attack, our method can further pose a challenge to the effectiveness of these defense mechanisms.
- **The impact of hyperparameters**: We have added new experiments to investigate the impact of the following parameters: the number of clusters, the number of classes in the dataset, and the accuracy of topology detection,
- **Scalability and cost on complicated topology**: We have added new experiments to report the performance and computational cost on topologies with more clients and random structures. Our method also works for DFL with complicated topology and the computational cost can be ignored compared with the training time of DFL.

For more detailed, point-by-point responses, please refer to our response to each reviewer.

We deeply appreciate the time and effort each of you has invested in this process. Please kindly let us know if you have any further questions, and we would be more than happy to resolve them before the rebuttal deadline.

---

### Meta-Review · Area_Chair_tv6f · 2024-12-24

**Metareview:**

The authors proposed distributed backdoor attacks in decentralized FL, introducing a method to detect attackers by estimating distances between them, clustering attackers accordingly, and proposing an algorithm to dynamically embed local patterns from a global pattern into each cluster. The novelties and contributions are quite limited for this work compared with the original DBA attack which is designed for FL, and this work lacks discussion on the key parameters of the proposed method in the experiment, such as the number of clusters.
The authors are encouraged to add more ablation studies to evaluate the contribution of each module to the attack success rate.

**Additional Comments On Reviewer Discussion:**

The reviewers are agreed on the final decision.

---

### Decision · Program_Chairs · 2025-01-22

Reject